# Development of a Prognostic Tool to Guide the Decision to Extend Adjuvant Aromatase Inhibitors for up to Ten Years in Postmenopausal Early Breast Cancer Patients

**DOI:** 10.3390/cancers12123725

**Published:** 2020-12-11

**Authors:** Camille Moreau-Bachelard, Loïc Campion, Marie Robert, Olivier Kerdraon, Céline Renaudeau, Maud Aumont, Jean-Marc Classe, Mario Campone, Jean-Sébastien Frénel

**Affiliations:** 1Department of Medical Oncology, Institut de Cancérologie de l’Ouest, Boulevard Professeur Jacques Monod, 44800 Saint-Herblain, France; camille.moreau-bachelard@ico.unicancer.fr (C.M.-B.); marie.robert@ico.unicancer.fr (M.R.); Mario.Campone@ico.unicancer.fr (M.C.); 2Department of Biometrics, Institut de Cancérologie de l’Ouest, Boulevard Professeur Jacques Monod, 44800 Saint-Herblain, France; loic.campion@ico.unicancer.fr; 3CRCINA, University of Nantes, INSERM UMR1232, CNRS-ERL6001, 44000 Nantes, France; 4Department of Pathology, Institut de Cancérologie de l’Ouest, Boulevard Professeur Jacques Monod, 44800 Saint-Herblain, France; olivier.kerdraon@ico.unicancer.fr; 5Department of Surgery, Institut de Cancérologie de l’Ouest, Boulevard Professeur Jacques Monod, 44800 Saint-Herblain, France; celine.renaudeau@ico.unicancer.fr (C.R.); jean-marc.classe@ico.unicancer.fr (J.-M.C.); 6Department of Radiation Oncology, Institut de Cancérologie de l’Ouest, Boulevard Professeur Jacques Monod, 44800 Saint-Herblain, France; maud.aumont@ico.unicancer.fr

**Keywords:** early breast cancer, hormone receptor, positive, late recurrence

## Abstract

**Simple Summary:**

Postmenopausal women with hormone receptor-positive early breast cancer receive adjuvant aromatase inhibitors (AIs) for five years. However, recurrences still occur at a steady rate over at least twenty years, and extending adjuvant AIs for up to ten years is an option. Our work focuses on a specific population of postmenopausal patients who have already received five years of adjuvant AIs. This population is at high risk of osteoporosis and extending AIs must be carefully decided in line with the potential benefit. In this study, we developed a simple tool to identify women at high risk of relapse despite completing five years of AI treatment. The score it provides is available free of charge upon diagnosis and divides patients into two prognostic groups. Combining that score with comorbidities, bone mineral density, and patient motivation may help the decision-making process to recommend extending adjuvant AIs.

**Abstract:**

*Background:* The selection of women with hormone receptor-positive (HR+) early breast cancer (EBC) at high risk of relapse after five years (yrs.) of adjuvant aromatase inhibitors (AIs) is crucial, as the benefit of extending AIs is counterbalanced by toxicity. We developed a clinicopathological tool to estimate the residual risk of relapse after five years of adjuvant AIs. *Methods:* The Institut de Cancérologie de l’Ouest (ICO) database was used to determine a prognostic score of post-five-year AI relapse. Cox regression models estimated our score’s prognostic performance. *Results:* In total, 1105 women were included. Median follow-up was 44 months (IQR = 21–70) post-AI treatment. From the Cox models, we designed a dichotomous prognostic score including the number of macrometastases, age (>70 yrs. vs. ≤70 yrs.), tumor size (≥T2 vs. not), and mitotic activity (≥2 vs. not). Overall, 77.5% of patients were classified as being at low risk and 22.5% at high risk of late recurrence. Low-risk patients had a five- to ten-year local or distant recurrence risk of 7.6% (95% CI, 5.4% to 10.6%) as compared with 26.9% (95% CI, 19.9% to 35.7%) for the high-risk roup. *Conclusion:* In this study, we developed a simple tool to identify women at high risk of relapse despite completing five years of AIs.

## 1. Introduction

Late recurrences (i.e., five or more years after diagnosis) account for half of Estrogen receptor-positive (ER+) early breast cancer (EBC) recurrences. Adjuvant chemotherapy and five years of endocrine therapy (ET) significantly reduce the risk of relapse, particularly in the first five years, while endocrine therapy remains beneficial after five years due to a carryover effect [1,2,3,4,5].

Two different strategies tackle the issue of late relapse. Firstly, extending the duration of tamoxifen for up to ten years improved premenopausal women’s disease-free survival (DFS) and overall survival (OS) in the ATLAS and aTTom trials [6,7]. The benefit of extended AIs in postmenopausal women pretreated with five years of AIs is debated, as mixed results have been published. Extending adjuvant AIs beyond five years reduces the occurrence of secondary breast tumors but has no or little impact on distant metastasis-free survival [8,9,10]. Secondly, adjuvant ET combined with two or three years of adjuvant CDK4/6 inhibitors is currently being investigated, with increased invasive DFS with abemaciclib [11]. However, the follow-up is very short and the impact on late relapse is yet unknown.

The selection of patients who may benefit from such therapeutic escalation is paramount, as the potential survival impact is counterbalanced by lower quality of life among potentially cured patients [12]. Several clinicopathological predictors are currently used as decision-making tools for adjuvant systemic treatment [13,14,15]. Predictors of late recurrence are not well defined [16]. In a patient-level meta-analysis, patients within each tumor size and nodal status exhibited a significant risk of recurrence after five years, with the risk increasing with larger tumor size and nodal status [3]. The Clinical Treatment Score post-5 years (CTS5) is an algorithm integrating tumor grade, age, tumor size, and number of lymph nodes to predict the risk of distant recurrence after five years [17]. This score was designed using the ATAC study patients and included a mix of patients previously treated with five years of tamoxifen or aromatase inhibitors [18]. Finally, prognostic tests based on the initial tumor biology have tried to identify patients at risk of late recurrence. However, neither the American Society of Clinical Oncology nor the National Comprehensive Cancer Network suggest using the results of these assays to guide decisions for patients who are disease-free after five years.

In this study, we aimed to pinpoint factors of late recurrence in postmenopausal women with ER+ EBC who have already completed five years of adjuvant AIs. We aimed to develop a dedicated prognostic tool facilitating the decision-making process on extending adjuvant AIs beyond five years.

## 2. Patients and Methods

### 2.1. BERENIS Database

Base d’Evaluation et de REcherche des Néoplasmes Infiltrants et in Situ (BERENIS) is a database including all patients treated for BC at the Institut de Cancérologie de l’Ouest (ICO) in Nantes, France.

This database gathers individual retrospective data from all consecutive patients, male or female, ≥18 years, having started an anti-cancer treatment for an early breast cancer in our institution. Patient-related data are collected, including patient demographic characteristics, pathology, and outcomes. Treatment strategies are also recorded, including chemotherapy, targeted agents, endocrine therapy (ET), and radiotherapy (RT). Tolerance and side effects are not reported in the database and KI67 was not recorded in the database.

In accordance with French regulations, the French data protection authority (CNIL, Commission Nationale de l’Informatique et des Libertés) authorized the database (authorization number 706825). ICO manages it in compliance with Good Clinical Practice. The study was conducted in accordance with the Declaration of Helsinki, and the protocol was approved by the Ethics Committee of Angers. CNIL (2019/48). authorization No. 706825. Non-opposition and consent letters were sent to patients.

### 2.2. Study Population and Objectives

All postmenopausal women who were diagnosed between 2003 and 2011, treated for ER+ (ER > 10% and/or PR > 10%) EBC, and completed at least 54 months (≈4.5 years) of adjuvant AIs were included. Postmenopausal women who had previously received tamoxifen, relapsed during adjuvant treatment, or were lost from follow-up during the first five years or shortly after the end of AIs were excluded. As our goal was to identify a population of patients who could benefit from extended ET, patients with an ER- BC relapse were also excluded. Patients needed to have a minimum follow-up of one year after the end of their adjuvant AIs. The primary objective was to pinpoint factors of late recurrence defined as ipsilateral, contralateral, in situ, invasive, or metastatic breast cancer. The secondary objective was to determine a score to predict late recurrence.

### 2.3. Statistical Analysis

The continuous variables were described by mean, standard deviation, median, and range. The qualitative variables were described by the frequency of their respective modalities. Relapse-free survival was defined as the time between the end of ET and either the date of first relapse or the date of the last follow-up visit without relapse. Relapse-free survival curves were calculated by the Kaplan–Meier method and compared between relevant groups through the log-rank test. Median follow-up was calculated using the reverse Kaplan–Meier method. At the univariate step, prognostic factors for relapse-free survival were assessed using the log-rank test or the univariate Cox test. Variables that had a *p*-value of significance <0.20 at the univariate step were introduced in the semiparametric multivariate Cox model. The multivariate model’s validity for the proportional risks was verified. We used the multiple imputation method to keep the power and use all 1105 patients (109 events) without bias. Moreover, we tested this imputed model’s robustness with a sensitivity analysis via maximum bias testing (either all missing data = 0 or 1). A continuous prognostic score was calculated from imputed model coefficients and applied to each of the 1037 complete cases. To obtain a binary classification score, optimal cut-off was assessed using the Contal and O’Quigley method, and the corresponding risk curves were traced.

All tests were two-sided and the significance limit was set at 5%. All analyses were performed using SAS 9.4 software (SAS Institute Inc., Cary, NC, USA) and Stata Special Edition 16.1 (StataCorp, College Station, TX, USA).

## 3. Results

### 3.1. Patient Characteristics and Management

Of the 1496 patients identified in the database, 1105 met the inclusion criteria. The main reasons for excluding patients were recurrence during ET (*n* = 10), HR- relapse (*n* = 10), and loss of follow-up immediately after the end of adjuvant ET (*n* = 371). The flow chart is shown in Appendix A.

The median follow-up from diagnosis was 111 months (IQR = 87–136). The median age at primary diagnosis was 62 years (IQR = 58–69). The patients’ clinical and pathological features are shown in Table 1. Tumor size was mainly pT1 (77.6%) or pT2 (20.6%) while the Elston–Ellis grade was I, II, and III in 29.6%, 52.8%, and 12.2% of patients, respectively. ER/PR status was distributed as follows: ER+/PR+ (76.9%), ER+/PR− (22.6%), ER−/PR+ (0.5%). HER2 status was positive in 6.5% and missing in 1.8% of cases. Regarding disease management, 71.6% of patients underwent conservative surgery associated with an axillary node clearance (62%) or axillary sentinel lymph node biopsy (38%). Nodal involvement was found in 26.1% of patients. Chemotherapy was given to 43% (474/1105) of patients, and 75% (54/72) of HER2+ BC patients received adjuvant trastuzumab. All patients received adjuvant AIs with a median duration of 60 months (IQR = 59–60). Anastrozole was mainly prescribed (57.4%).

### 3.2. Pattern and Management of Patients Who Relapsed after Five Years of AIs

Median follow-up was 44 months after AI completion (IQR = 21–70). The recurrence rate was 9.9% (*n* = 109) with a median time to relapse of 31 months (17–51). These patients’ characteristics are described in Table 2. Median age at relapse was 65 years (IQR = 60–72). Distant relapse occurred in 65.1% (*n* = 71) of cases, and relapse was contralateral in 26.6% (*n* = 29). Carcinoma in situ represented 6.4% of local relapse. At the end of data collection, all patients with ipsilateral relapse were alive, but one presented with subsequent metastatic progression. For patients with contralateral relapse, two died due to their metastatic disease. Among 71 patients with metastatic relapse, 52.1% had more than one metastatic site, including bone (71.8%), liver (22.5%), and pleura (21.1%). First-line treatment of metastatic disease included ET alone (*n* = 38; 53.5%), ET and targeted therapy (*n* = 17; 23.9%), chemotherapy (*n* = 11; 15.5%), and best supportive care (*n* = 5; 7.1%). The median follow-up of patients after metastatic relapse was 22 months.

### 3.3. Clinical Factors Associated with Recurrence after Five Years of AIs

We conducted a univariate analysis to identify the factors associated with late relapse. Age >70 years at initial diagnosis, tumor size, nodal involvement, lymphovascular invasion, lobular subtype, and neo- or adjuvant chemotherapy were associated with a higher risk of late relapse. The Elston–Ellis grade was generally not associated with a higher risk of late relapse. However, tubule formation and mitotic activity, two components of the Elston–Ellis grade, were significant in this univariate analysis. The details are shown in Table 3. Conversely, HER2 positivity, ER+/PR− or ER−/PR+, were not associated with an increased risk of late recurrence. In the multivariate analysis, tumor size (>pT1), degree of node involvement, age >70 years, and mitotic activity, which is a component of the grade (2/3 versus 1), remained independent prognostic factors of recurrence after completing adjuvant AIs. The data are detailed in Table 3.

### 3.4. Integration of Clinical Variables to Develop a Prognostic Tool

We aimed to develop a simple prognostic tool to estimate the risk of late recurrence based on clinicopathological parameters measured virtually in all patients with BC at diagnosis. We first identified prognostic variables and then devised a raw score to identify different prognostic groups in terms of disease-free survival after five years of AIs. To develop this score, points were attributed to each independent prognostic variable. The numbers of points given to each one were determined from its beta coefficient in the final Cox model: 100 points for each macrometastasis, 383 points if the age was over 70 years at diagnosis, 446 points if the tumor size was more than T1, and 499 points if the mitotic activity was 2 or 3. Then, according to each patient’s prognostic variable status, the raw score was calculated as the total corresponding points.

In the complete cases population, the median score was 200 (range = 0–2728). The optimal cut-off for good versus poor prognosis (*n* = 500) was calculated using the Contal and O’Quigley method (Figure 1). Overall, 77.5% of patients were classified as being at low risk and 22.5% at high risk of late recurrence. The characteristics of patients in the low- and high-risk groups are shown in Table 4. As expected, most but not all low-risk patients were node negative, and interestingly, 30.9% of high-risk patients were node negative. The low-risk group had a mean five- to ten-year recurrence risk of 7.6% (95% CI, 5.4% to 10.6%) as compared with 26.9% (95% CI, 19.9% to 35.7%) for the high-risk group. The probability of disease-free survival according to low- and high-risk score is shown in Figure 2.

Interestingly, as show in Figure 2, the two populations (low and high risk) separate 9 to 12 months after the end of adjuvant ET. This corresponds to the second peak of relapse that occurs around 7 years after the diagnosis of breast cancer.

## 4. Discussion

Estimating the risk of late recurrence is a major challenge to optimize the management of women with ER + BC. In this study, we developed a simple clinicopathological score that is able to identify postmenopausal women at risk of late relapse despite completing five years of AI treatment. By integrating tumor size, nodal involvement, age, and mitotic activity, our score classifies patients as being at low risk of local or distant recurrence five to ten years after diagnosis—7.6% (95% CI, 5.4% to 10.6%)—or at higher risk—26.9% (95% CI, 19.9% to 35.7%). These clinicopathological variables are easily available for all patients at diagnosis. That makes this score readily available to all clinicians.

The originality of this study is in the development of a prognostic score for late relapse in a population of women all pretreated with five years of adjuvant AIs. The question of extending adjuvant AIs in this context is a frequent clinical situation. Extended adjuvant endocrine therapy can reduce the risk of breast cancer recurrence and continue to reduce the risk of contralateral breast cancer [5], [19,20,21,22]. However, the greatest impact of such a strategy is seen for women who have received five years of tamoxifen and who will continue for five more years or switch to an AI [6,7,19,20,22]. The benefit of extended AIs in women pretreated with five years of AIs is debated, as mixed results have been published [5]. The MA.17R trial was the first to randomize patients with up to ten years of AIs. After a median follow-up of 6.3 years, DFS was significantly reduced with a hazard ratio (HR) of 0.66 (*p* = 0.01) [20]. The AERAS and ABCSG 6a trials have confirmed this DFS benefit [19,21]. Conversely, the NSABPB-42 trial has shown that letrozole does not extend DFS in disease-free patients after five years of AIs [8]. The DATA, LATER, ABCSG16, and IDEAL trials have also failed to demonstrate any DFS and OS benefit [9,23,24,25]. In 2018, however, a meta-analysis from the Early Breast Cancer Trialists’ Collaborative Group (EBCTCG), including eleven clinical trials, suggested a 20% reduced risk of relapse (HR 0.81; 95% CI 0.73–0.90) with five years of extended AIs in women pretreated with AIs [5]. Recently, a meta-analysis of eight trials investigating AI extension has demonstrated a significant improvement in DFS (relative risk (RR) 0.79; 95% CI 0.68–0.91), distant recurrence (RR 0.75; 95% CI 0.58–0.96), and contralateral breast cancer (RR 0.53; 95% CI 0.40–0.70) [26]. Patients with tumor size >2 cm, nodal involvement, and previous chemotherapy derived the most DFS benefit.

The accurate selection of patients who derive the most benefit from extended AIs is key. High-risk women with our score have a persistent risk of relapse of up to 26.6% between years five to ten, making them good candidates for extended endocrine therapy. Other algorithms have been published recently. The CTS5 score was developed in postmenopausal women in the ATAC trial and validated in the BIG 1–98 study [17,18,27]. It divides patients into three groups of risk of distant recurrence (DR): low (<5% DR risk, years five to ten), intermediate (5% to 10%), or high (>10%). In contrast to our study, this score included a mix of patients treated with either five years of tamoxifen (50.1%) or five years of AIs (49.9%). As the mechanisms of resistance to these two drugs are dissimilar, we think that studies such as ours, focusing on women who have been treated with five years of AIs alone, are of value. We also have some additional differences to the CTS5 score. Firstly, our score is dichotomized and may be of greater help to clinicians as no intermediate group is defined. Secondly, our low-risk group harbors a 7.6% residual risk of relapse compared to <5% in the CTS5 score, which may be considered quite high. We emphasize that our score includes local relapse, as adjuvant endocrine therapy not only prevents metastatic recurrence but also reduces local–regional relapse by 40% and contralateral relapse by 50% [2]. Finally, none of the patients included in our study have benefited from extended adjuvant ET. The CTS5 score is based on the ATAC and BIG 1–98 populations, and none of these trials have collected comprehensive information on the use of extended adjuvant endocrine therapy. Indeed, some patients in these trials may have benefited from extended adjuvant ET, which could influence the conclusion.

Our score of late relapse integrates mitotic activity, which is a proliferating biomarker, often associated with early recurrence in HR+ breast cancer [28]. Some studies have already demonstrated the association between late recurrence and proliferating biomarkers. The KI-67 labelling index (LI) is a proliferating biomarker, associated with mitotic activity [29]. In a cohort of lobular carcinoma, Ki-67 LI was associated with late distant recurrence [30]. More recently, Ki-67 LI (>20%) was associated with a significant risk of distant relapse despite a low-risk CTS5 score [31]. The authors stated that endocrine therapy could be considered in patients with high Ki-67 LI (>20%) in the low CTS5 group and that the combination of CTS5 and KI-67 could predict more accurately the risk of late relapse. Ki-67 LI was not recorded in our database and was, therefore, not included in our model.

Beyond the conventional clinicopathological factors included in our score or in CTS5, molecular signature may help to identify patients at risk of late relapse. These tests include the immunohistochemical 4 (IHC4) protein test, 21-gene Recurrence Score (Oncotype DX), PAM50 intrinsic subtype (Prosigna), 12-gene Recurrence Score (EndoPredict), two-component Breast Cancer Index (based on the molecular grade index and HOXB13: IL17BR), and 70-gene signature (MammaPrint) [14,24,32,33,34,35,36,37,38]. They are performed routinely for patient prognosis and to facilitate the decision-making process on adjuvant chemotherapy delivery. The tests’ ability to predict post-five-year prognosis has only been evaluated in a prospective retrospective manner. Of these assays, only the two-component Breast Cancer Index was shown to be predictive of benefit from letrozole after five previous years of tamoxifen [39]. Validation studies are lacking and, considering the level of evidence, no oncology society recommends the assays to guide the extension of adjuvant endocrine therapy. Beyond these molecular signatures, only few data are available about the genomic characteristics of late recurrent tumors. In an analysis of the SOLE phase 3 trial which evaluated the effect of extended intermittent or continuous letrozole on late relapse in postmenopausal women with hormone receptor-positive early breast cancer, *FGFR1* copy number gain was significantly associated with an increased risk of late breast cancer events in univariate and multivariable models adjusted for clinicopathologic factors [40]. ESR1 mutations, which are frequent and associated with resistance to AIs in the metastatic setting, are very rare events (0.5%) in primary tumors and have not been associated so far with late relapse [41].

Ultimately, clinicopathological factors remain the most used tool to consider extending adjuvant AIs or not. However, patient comorbidities and preferences play a key role in the decision. Arthralgia, myalgia, bone loss, and cardiovascular events have been reported frequently during AI use. The percentages of discontinuation range from 11.7% during the first year to 31.3% at 5 years during adjuvant AIs [42]. Early discontinuation rates in the trials investigating extended endocrine therapy are as high as 30%. We emphasize that our population is a selection of women who had clinically tolerated 5 years of AIs and had presumably few arthralgia and myalgia. The prolongation of AIs induced and increased odds of cardiovascular events (OR = 1.18, 95% CI = 1.00 to 1.40, *p* = 0.05), bone fractures (OR = 1.34, 95% CI = 1.16 to 1.55, *p* < 0.001) in a meta-analysis of seven clinical trials [43,44]. These latter side effects should be taken into consideration in patients with preexisting comorbidities or risk factors and deserve specific management.

Our study has several limitations. The retrospective design may have induced some selection bias. However, the ICO BERENIS database included all consecutive patients from our institute. Secondly, validation in an independent population would be required to implement the score more routinely in practice. Finally, patient compliance and tolerance cannot be evaluated and may impact the outcome. Several studies have highlighted patients’ low rates of compliance with adjuvant ET and extending this treatment would involve only very motivated patients [45]. Nonadherence to treatment is underrecognized by physicians and significantly impairs the outcome, as demonstrated recently. Therapeutic drug monitoring is not routinely performed and may be a solution.

## 5. Conclusions

Extending adjuvant AIs after five years of treatment is a therapeutic option given the impact on DFS, as demonstrated in the EBCTCG meta-analysis. An accurate selection of patients who might derive the most benefit is crucial given the side effects associated with AIs. We have developed a simple tool to identify postmenopausal women at high risk of relapse despite completing five years of AIs. The score it provides is available free of charge upon diagnosis and divides patients into two prognostic groups. Combining this score with comorbidities, bone mineral density, and patient motivation may help the decision-making process to recommend extending adjuvant AIs.

## Figures and Tables

**Figure 1 cancers-12-03725-f001:**
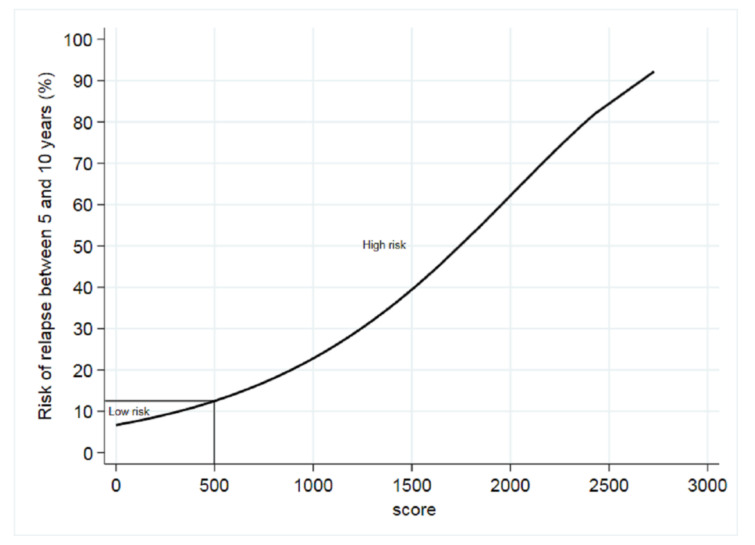
Risk of relapse between five and ten years according to the score. It corresponds to the cumulative risk of local–regional and metastatic relapse in low-risk patients five to ten years after diagnosis. We represented the risk of relapse-free survival after the end of endocrine therapy (ET) according to the score. The solid vertical line indicates the cut-off point for at-risk groups.

**Figure 2 cancers-12-03725-f002:**
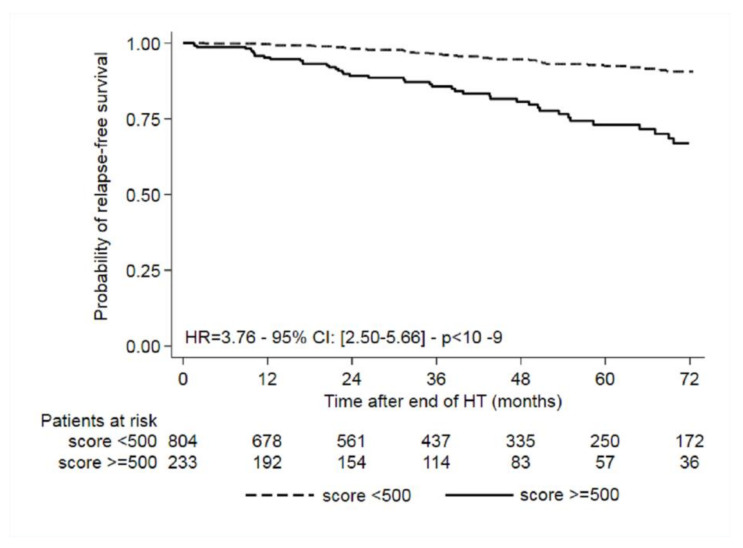
Probability of relapse-free survival according to low- and high-risk score.

**Table 1 cancers-12-03725-t001:** Characteristics of the population (*n* = 1105).

Characteristics	*N*	%
**Age**		
Mean (years)	63.3	
Median (range)	63 (33–88)	
<50 yrs.	11	1
50–70 yrs.	897	81.2
>70 yrs.	197	17.8
**Tumor size**		
pT0	3	0.3
pT1	858	77.6
pT2	228	20.6
pT3	16	1.4
**Nodal status**		
pN0	817	73.9
pN1	218	19.7
pN2	57	5.2
pN3	13	1.2
**Micrometastasis**		
Yes	97	8.8
No	1010	91.2
**Hormone Receptor**		
Estrogen Receptor +	1100	99.5
Estrogen Receptor −	5	0.5
Progesterone Receptor +	855	77.4
Progesterone Receptor −	250	22.6
**HER2 status**		
Positive	72	6.5
Negative	988	89.4
**Histological subtype**		
Ductal carcinoma	839	76
Lobular carcinoma	197	17.8
Mix	27	2.4
Other	42	3.8
**Elston–Ellis grade**		
I	327	29.6
II	584	52.8
III	135	12.2
**Tubule formation**		
1	126	11.4
2	331	30
3	579	52.4
**Nuclear polymorphism**		
1	14	1.3
2	733	66.3
3	289	26.2
**Mitotic activity**		
1	801	72.5
2	158	14.3
3	78	7.1
**Emboli**		
Yes	165	14.9
No	927	83.9
**Surgery**		
Radical	313	28.3
Conservative	791	71.6
**Chemotherapy**		
Neoadjuvant	77	6.9
Adjuvant	397	35.9
No	632	57.2
**Chemotherapy drug**		
Anthracycline	442	93.4
Cyclophosphamide	457	96.6
Taxane	370	78.2
Clinical trial	50	10.6
**Endocrine therapy drug**		
Anastrozole	634	57.4
Letrozole	284	25.7
Exemestane	187	16.9

**Table 2 cancers-12-03725-t002:** Characteristics of relapsing patients (*n* = 109).

Characteristics of Relapses	*N*	%
**Age**		
Mean (years)	62.7	
Median (range)	62 (39–83)	
<50 yrs.	1	1
50–70 yrs.	88	80.7
>70 yrs.	20	18.3
**Location**		
Locoregional	9	8.3
Contralateral	29	26.6
Metastasis	71	65.1
**Histology**		
Invasive ductal carcinoma	69	63.3
Invasive lobular carcinoma	15	14
In situ	7	6.4
**Time to relapse after five years of AIs**		
Median (range)	31 (17–51)	
**First metastasis location**		
**Single**	*n* = 34	47.9
Bone	16	
Pleural	6	
Hepatic	4	
Cutaneous	3	
Brain and SNC	2	
Pulmonary	1	
Osteomedullary invasion	1	
Lymph node involvement	1	
**Multiple, including:**	*n* = 37	52.1
Bone	25	
Hepatic	9	
Lymph node involvement	9	
Pleural	8	
Pulmonary	8	
Cutaneous	6	
Brain and SNC	3	
Peritoneal	3	
Ovarian	1	
Osteomedullary invasion	1	
**Management for metastatic relapse**		
Chemotherapy	11	15.5
Hormonotherapy alone	38	53.5
Hormonotherapy and metabolic targeted therapy	17	23.9
Best supportive care only	5	7.1

**Table 3 cancers-12-03725-t003:** Univariate and multivariate analysis of factors associated with late relapse.

Characteristics	Univariate Analysis				Multivariate Analysis			
	Hazard Ratio	*p* > |z|	95% CI		Hazard Ratio	*p* > |z|	95% CI	
Age ≥ 70 yrs. vs. <70 yrs.	**1.64**	**0.036**	**1.03**	**2.61**	**1.68**	**0.033**	**1.04**	**2.69**
Age (continuous)	1.01	0.382	0.99	1.04				
Number of macrometastases (continuous)	**1.18**	**0.000**	**1.13**	**1.23**	**1.13**	**0.000**	**1.08**	**1.20**
Tumor size (continuous)	**1.02**	**0.000**	**1.02**	**1.03**				
pT 2–3 vs. 0–1	**2.77**	**0.000**	**1.89**	**4.05**	**1.58**	**0.060**	**0.98**	**2.55**
Tumor grade II vs. I	1.27	0.327	0.79	2.06				
Tumor grade III vs. I	1.90	0.039	1.03	3.51				
Tumor grade III vs. I–II	1.62	0.065	0.97	2.71				
Nuclear pleomorphism	0.94	0.799	0.59	1.50				
Tubule formation 2–3 vs. 1	**4.05**	**0.017**	**1.28**	**12.80**	2.28	0.18	0.68	7.68
Mitotic activity 2–3 vs. 1	**2.43**	**0.000**	**1.60.**	**3.68**	**1.89**	**0.009**	**1.18**	**3.05**
Number of micrometastases	1.54	0.11	0.91	2.63				
ER and PR dissociation vs. no	0.93	0.76	0.60	1.45				
HER2 overexpressed vs. no	0.75	0.49	0.33	1.71				
Emboli 0 vs. positive	**1.92**	**0.003**	**1.25**	**2.94**	1.15	0.564	0.72	1.84
Lobular histology vs. other	**1.60**	**0.033**	**1.04**	**2.48**	1.36	0.225	0.83	2.22
Lobular or ductal histology vs. other	1.37	0.152	0.89	2.10				
Chemotherapy vs. no	**2.24**	**0.000**	**1.52**	**3.31**	1.06	0.817	0.65	1.74
Radiotherapy vs. no	1.29	0.61	0.47	3.52				

Characteristics that are clinically significant are highlighted. Only some variables from the same category were tested in the multivariate analysis. Variables that had a *p*-value of significance < 0.20 in the univariate analysis were introduced in the semi-parametric multivariate Cox model.

**Table 4 cancers-12-03725-t004:** Characteristics of the low- and high-risk populations defined by our prognostic score.

Variable	Low Risk < 500		High Risk > 500		*p*-Value
	*N* = 804	%	*N* = 233	%	
**Mean age at tumor diagnosis (years)**	62.3 (6.7)		65.4 (9.1)		<10^−6^
**pT**					
0–1	739	91.9	91	39	<10^−6^
2–3	65	8.1	142	60.9	
**pN**					
N0	710	88.3	72	30.9	<10^−6^
N1	90	11.2	103	44.2	
N2	4	0.5	47	20.2	
N3	0	-	11	4.7	
**Micrometastasis**					
Yes	64	8	23	9.9	0.354
No	740	92	210	90.1	
**Estrogen receptor**					
Positive	800	99.5	232	99.6	0.895
Negative	4	0.5	1	0.4	
**Progesterone receptor**					
Positive	625	77.7	183	78.5	0.794
Negative	179	22.3	50	21.4	
**HER2 status**					
Positive	41	5.1	24	10.3	0.004
Negative	730	90.8	199	85.4	
**Histology**					
Invasive lobular carcinoma	126	15.7	55	23.6	0.023
Invasive ductal carcinoma	620	77.1	169	72.5	
Mix	20	4.5	6	2.6	
Other	38	4.7	3	1.3	
**Tumor grade**					
I	312	38.8	13	5.6	<10^−6^
II	445	55.3	135	57.9	
III	47	5.8	85	36.5	
**Tubule formation**					
1	123	15.3	3	1.3	<10^−6^
2	266	33.1	65	27.9	
3	415	51.6	163	70	
**Nuclear pleomorphism**					
1	14	1.7	0	0	<10^−6^
2	611	76	122	52.4	
3	179	22.3	110	47.2	
**Mitotic activity**					
1	716	89.1	85	36.5	<10^−6^
2	63	7.8	95	40.8	
3	25	3.1	53	22.7	
**Emboli**					
Yea	77	9.6	72	30.9	<10^−6^
No	722	89.8	160	68.7	
**Surgery**					
Radical	165	20.5	109	46.8	<10^−6^
Conservative	639	79.5	124	53.2	
**Adjuvant chemotherapy**					
No	579	72	41	17.6	<10^−6^
Yes	225	28	192	82.4	

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
