# Peer review of "Development of a Prognostic Tool to Guide the Decision to Extend Adjuvant Aromatase Inhibitors for up to Ten Years in Postmenopausal Early Breast Cancer Patients"

_cancers, 2020, doi:10.3390/cancers12123725_

Round 1

Reviewer 1 Report

The major concern of this study is the lack of analysis on Ki-67 and ESR1 mutation. Regarding the prognostic factors, other than the clinical and pathological features reported in this manuscript, Ki-67 and ESR1 mutations have been investigated intensively in many other studies. For example, Br J Cancer. 2020 Mar;122(6):759-765, J. Natl. Cancer Inst. 2007;99, 167–170; J. Clin. Oncol. 2005;23, 2477–2492, Biochim Biophys Acta Rev Cancer. 2019 Dec;1872(2):188315. Therefore, it is suggested that the authors should address the rationale for not including Ki-67 and ESR1 mutations.

A minor issue is explicity of exclusion process. Among the 1496 patients identified, 1105 met the inclusion criteria. The details of exclusion reasons and corresponding patient numbers should be provided. An algorithm of cohort selection is suggested.

Author Response

Response to Reviewer 1 Comments

Point 1: The major concern of this study is the lack of analysis on Ki-67 and ESR1 mutation. Regarding the prognostic factors, other than the clinical and pathological features reported in this manuscript, Ki-67 and ESR1 mutations have been investigated intensively in many other studies. For example, Br J Cancer. 2020 Mar;122(6):759-765, J. Natl. Cancer Inst. 2007;99, 167–170; J. Clin. Oncol. 2005;23, 2477–2492, Biochim Biophys Acta Rev Cancer. 2019 Dec;1872(2):188315. Therefore, it is suggested that the authors should address the rationale for not including Ki-67 and ESR1 mutations.

  • We thank the reviewer 1 for this important comment. In France, Ki-67 LI in not routinely performed for breast cancer and was not available in our database. We acknowledge that Ki67 LI has been recently associated also with late relapse and we have updated the bibliography to provide to the readers the most comprehensive view. We have also pointed the link between the mitotic activity (a component of our score) and Ki67 LI.

        In section 2.1 (page 2, line 37 - 42) we specified inclusion criteria and data collected. Indeed, The Ki67 was not recorded in the database. “This database gathers individual retrospective data from all consecutive patients, male or female, 18 years, having started an anti-cancer treatment for an early breast cancer in our institution. Patient-related data are collected, including patient demographic characteristics, pathology, and outcomes. Treatment strategies are also recorded, including chemotherapy, targeted agents, endocrine therapy (ET), radiotherapy (RT). Tolerance and side effects are not reported in the database and KI67 was not recorded in the database.”

        In the fourth paragraph of the discussion (page 11, line 17 – 25), we discussed the correlation between mitotic activity and Ki-67. “Our score of late relapse integrates mitotic activity that is a proliferating biomarker, often associated with early recurrence in HR+ breast cancer [28]. Some studies have already demonstrated the association between late recurrence and proliferating biomarkers. KI-67 labelling index (LI) is a proliferating biomarker, associated with mitotic activity [29]. In a cohort of lobular carcinoma, Ki-67 LI was associated with late distant recurrence [30]. More recently, Ki-67 LI (> 20%) was associated with a significant risk of distant relapse despite a low risk CTS5 score [31]. The authors stated that endocrine therapy could be considered in patients with high Ki-67 LI (> 20%) in the low CTS5 group and that the combination of CTS5 and KI-67 could predict more accurately the risk of late relapse. Ki-67 LI was not recorded in our database and was therefore not included in our model.”

  • Regarding ESR1 mutation, ESR1 mutations have been shown to be very rare events in endocrine naïve early breast tumors. As our goal was to provide a tool available at initial diagnosis, we did not look at ESR1 mutation in tumors. We have added a sentence in the manuscript to explain this.

For ESR1 mutations: In the fifth paragraph of the discussion (page 11, line 37 – 45), we explain why we did not research the ESR1 mutation. Beyond these molecular signatures, only few data are available about the genomic characteristics of late recurrent tumors. In an analysis of the SOLE phase 3 trial who evaluated the effect of extended intermittent or continuous letrozole on late relapse in postmenopausal women with hormone receptor-positive early breast cancer, FGFR1 copy number gain was significantly associated with an increased risk of late breast cancer events in univariate and multivariable models adjusted for clinicopathologic factors[40]. ESR1 mutations who are frequent and associated with resistance to AIs in the metastatic setting,  are very rare events (0.5%) in primary tumors and have not been so fare associated with late relapse [41].” 

 .

Point 2: A minor issue is explicity of exclusion process. Among the 1496 patients identified, 1105 met the inclusion criteria. The details of exclusion reasons and corresponding patient numbers should be provided. An algorithm of cohort selection is suggested.

  • Thank you for this suggestion. We have modified the manuscript.

        We specified the exclusion process in results section 3.1. and in a flow chart in supplemental 1. (Page 3, line 33 - 35) “The main reasons for excluding patients were recurrence during ET (n=10), HR- relapse (n=10) and loss of follow-up immediately after the end of adjuvant ET (n=371). The flow chart is shown in Supplemental 1.”

Supplemental 1 : Flow chart

Thank you for your consideration for this manuscript.

Sincerely,

Dr Camille Moreau-Bachelard & Dr Jean-Sébastien Frénel 

Reviewer 2 Report

The paper adresseses a  debated topic in the literature as recent meta-analyses show [1-3] with different results. None of them demonstrate a benefit on overall survival and only two have a positive effect on the disease free survival [1-2]. The criteria for continuing therapy beyond 5 years emerge only from   one paper [2]: tumour size > 2cm, node positive status and previous chemotherapy, while in the other two the evaluation if continue is only generically in "selected women with high risk tumour factors" [3]. Therefore the paper by Moreau-Bachelard et coll. has the meritorious purpose of offering a score for the decision making on a very controversial topic. Furthermore. the score is correctly constructed on a cohort of patients treated with AIs only, (and not with tamoxifen +AIs)  therefore able to benefit more from the prolongation of therapy [4].

However, here are few comments and questions for the authors:

1.In figure 2 relating to the Kaplan-Meier probability of relapse-free survival, the curves diverge after 12 month. Can the authors provide an explanation in the discussion?

2.Surely the score is easy-to-use. Can the authors provide more details on the prevalence, in the cohort, of comorbidities and adverse events of AIs as bone fractures, osteoporosis, arthralgia, joint stifness and hot flashes?. Furthermore it would be useful in the discussion to report the authors's experience relating to management of adverse events on the decision or not to continue therapy. In others words, should they be considered or not in the decision of the prolongation or must be used only the score?. Do they all have the same weight in decison making or some are more important than others?. This aspect is clinically relevant. Also the guidelines are not completely useful highlighting only that "shared decision making between clinicians and patients is appropriate". [5}

References:

1.Lin XU et al. Extendede adjuvant therapy with Aromatase inhibitors for early brest cancer: a meta-analysis of RCTs. Clim Breast Cancer: 2019 oct; 19 (5): e578-e588.

2.Xiaoying Qian et al. Effiacy and toxicity of extende aromatase inhibitor after adjuvant aromatase inhibitor-containg therapy for hormone-receptor-positive breast cancer: a literature-based meta-anlaysis of RCTs. Breast Cancer Res. 2020 Jan,  179 (2); 275-285.

3.Clement Z et al. Extended duration of adjuvant aromatase inhibitor in breast cancer: a meta-analysis of RCTs. Gland  Surg. 2018 Oct; 7 (5): 449-457.

4. Goss PE et al. Extending aromatase-inhibitor adjuvant therapy to 10 years. N Engl Med 2016; 375: 209-219.

5.Burstein et al. Adjuvant endocrine therapy for women with hormone receptor-positive Breast cancer: ASCO clinical practice guideline focused update. J Clin Oncol. 2019 Feb 10; 37 (5): 423-438.

Author Response

Response to Reviewer 2 Comments:

Point 1: In figure 2 relating to the Kaplan-Meier probability of relapse-free survival, the curves diverge after 12 month. Can the authors provide an explanation in the discussion?

  • We thank the reviewer 2 for this important comment. It is intriguing that the curves separate around 1 year after the end of adjuvant AIs. We believe that this time point corresponds to the 7th years after the diagnosis of breast cancer and correspond to the 2nd peak of relapse in HR+ breast cancer. We have modified the text to add this precision.

               (page 7, line 22 – 24) “Interestingly, as show on the Figure 2, the two populations (low and high risk) separate 9 to 12 months after the end of adjuvant ET. This corresponds to the second peak of relapse that occurs around 7 years after the diagnosis of breast cancer.”

Point 2: Surely the score is easy-to-use. Can the authors provide more details on the prevalence, in the cohort, of comorbidities and adverse events of AIs as bone fractures, osteoporosis, arthralgia, joint stifness and hot flashes?. Furthermore it would be useful in the discussion to report the authors's experience relating to management of adverse events on the decision or not to continue therapy. In others words, should they be considered or not in the decision of the prolongation or must be used only the score? Do they all have the same weight in decison making or some are more important than others?. This aspect is clinically relevant. Also the guidelines are not completely useful highlighting only that "shared decision making between clinicians and patients is appropriate".

  • As pointed by the reviewer 2, comorbidities are very important to consider in the patients. In our database, these data were not collected and therefore not available for this manuscript. We acknowledge that our population of patients have clinically tolerated 5 years of adjuvant AIS and are a selection of patients. This comment has been added in the manuscript to make the readers aware of this potential selection bias. In addition, the discussion section has been improved enriched on the topic of side effects and comorbidities.

               Firstly, in patients and methods, section 2.1 (page 2, line 37 to 42) we specified inclusion criteria and data collected. Indeed, tolerance and side effects were not reported in the database.This database gathers individual retrospective data from all consecutive patients, male or female, 18 years, having started an anti-cancer treatment for an early breast cancer in our institution. Patient-related data are collected, including patient demographic characteristics, pathology, and outcomes. Treatment strategies are also recorded, including chemotherapy, targeted agents, endocrine therapy (ET), radiotherapy (RT). Tolerance and side effects are not reported in the database and KI67 was not recorded in the database.”

Secondly, in discussion, paragraph 6, (page 11, line 48 – page 12, line 6), we added the prevalence of adverse events and discussed why we considered our patients as good observers with few adverse events. “Ultimately, clinicopathological factors remain the most used tool to consider extending adjuvant AIs or not. However, patient comorbidities and preferences play a key role in the decision. Arthralgia, myalgia, bone loss and cardiovascular events have been reported frequently during AIs use. The percentages of discontinuation range from 11.7% during the first year to 31.3% at 5 years during adjuvant AIs [42]. Early discontinuation rates in the trials investigating extended endocrine therapy are as high as 30%. We emphasize that our population is a selection of women who had clinically tolerated 5 years of AIs and had presumably few arthralgia and myalgia. The prolongation of AIs induced  and increased odds of cardiovascular events (OR = 1.18, 95% CI = 1.00 to 1.40, P = .05), bone fractures (OR =1.34, 95% CI = 1.16 to 1.55, P < .001) in a meta-analysis of seven clinical [43–44]. These latter side effects should be taken into consideration in patients with preexisting comorbidities or risk factors and deserve a specific management.”

               Finally, in paragraph 7, (page 12, line 12), we added and toleranceto the sentence: “Finally, patient compliance and tolerance cannot be evaluated and may impact the outcome.”

Thank you for your consideration for this manuscript.

Sincerely,

Dr Camille Moreau-Bachelard & Dr Jean-Sébastien Frénel
